# Recommender Forest for Efficient Retrieval

**Chao Feng[1]\*, Wuchao Li[1]\*, Defu Lian[1], Zheng Liu[2], Enhong Chen[1]**
[1]School of Computer Science and Technology
University of Science and Technology of China, Hefei, China
[2] Microsoft Research Asia, Beijing, China
{chaofeng,liwuchao}@mail.ustc.edu.cn
{liandefu,cheneh}@ustc.edu.cn, zhengliu@microsoft.com

## Abstract

Recommender systems (RS) have to select the top-n items from a massive item set. For the sake of efficient recommendation, RS usually represents users and items as latent embeddings and relies on approximate nearest neighbor search (ANNs) to retrieve the recommendation results. Despite the reduction of running time, the representation learning is independent of ANNs index construction; thus, the two operations can be incompatible, which results in a potential loss of recommendation accuracy. To overcome the above problem, we propose the Recommender Forest (a.k.a., RecForest), which jointly learns latent embedding and index for an efficient and high-fidelity recommendation. RecForest consists of multiple K-ary trees, each of which is a partition of the item set via hierarchical balanced clustering such that each item is uniquely represented by a path from the root to a leaf. Given such a data structure, an encoder-decoder-based routing network is developed: it first encodes user information into user representation; then, leveraging a transformer-based decoder, it identifies the top-n items via beam search. Compared with the existing methods, RecForest brings in the following advantages: 1) the false partition of the near-boundary items can be effectively alleviated by the use of multiple trees; 2) the routing operation becomes much more accurate thanks to the powerful transformer decoder; 3) the branch parameters are shared across different tree levels, making the index to be extremely memory-efficient. The experimental studies are performed on six popular recommendation datasets: with a significantly simplified training cost, RecForest outperforms competitive baseline approaches in terms of both recommendation accuracy and efficiency. The code is available at https://github.com/wuchao-li/RecForest.

## 1  Introduction

Recommender system (RS) is an important way to address the information overload problem. A typical recommender system needs to select the top-n items for users from a massive-scale item set. For the sake of efficient recommendation, RS usually calls for the collaboration of representation learning and Approximate Nearest Neighbour search (ANNs). In the first place, users and items are represented by embeddings in the same latent space; in the second place, the item embeddings are organized with a specific ANNs index, like SCANN and HNSW, such that the top-n recommendation to the user can be efficiently accomplished. Despite that the recommendation process is greatly accelerated from the above workflow, the recommendation quality will probably be restricted, given that the representation model is independently learned and can be incompatible with the ANNs index.

In recent years, many efforts have been devoted to alleviating the incompatibility between the representation model and the ANNs index, especially the effort on joint optimization of both components.

---

\*These authors contributed equally to this work; Defu Lian is the corresponding author.

36th Conference on Neural Information Processing Systems (NeurIPS 2022).

One representative class of works is the tree-based deep model (TDM) [29] and the joint optimized tree-based model (JTM) [28] from Alibaba. In both works, the item set is organized with a binary tree structure: each internal node acts like a cluster center, and each leaf node corresponds to a unique item. On top of the tree structure, a preference model is learned to route from the root to the leaf nodes for the top-n recommendation results. These works achieve empirical gains over the conventional two-stage methods; besides, they preserve competitive retrieval efficiency as the time cost is logarithmic to the size of the item set.

However, the existing tree-based recommenders are still restricted in many aspects. Firstly, the item set is hierarchically partitioned; as a result, it is challenging to route to items located around partition boundaries. Secondly, a routing decision is made without consideration of the routing trajectory (i.e., from the root to the direct ancestor of the current node), so the accuracy of beam search can be limited. Thirdly, the tree-based index can be memory-consuming, given that the number of internal nodes is at the same magnitude as the leaf nodes (i.e., the number of the items). Last but not least, the existing methods call for joint adaptation of the representation model and tree index, so that the tree structure needs to be repetitively updated, resulting in a significant cost for the training stage.

To overcome the limitations, we propose a novel framework, Recommender Forest (a.k.a., RecForest) for an efficient and high-fidelity recommendation. RecForest is highlighted for the following features.

- RecForest consists of multiple K-ary trees, each of which is a partition of the item set based on balanced hierarchical clustering. With the construction, the retrieval of the near-boundary items can be effectively improved, as evidenced in Section 3.6.1, since a near-boundary item missed in one tree can be retrieved back on another tree.

- RecForest leverages a transformer decoder for beam search. On top of such a decoder, the routing trajectory, i.e., from the root node to the current node, can be jointly considered when the next routing decision is made. Compared with the previous methods which merely take account of the current node, the beam search becomes much more accurate since the routing trajectory is fully utilized, as analyzed in Section 3.4.

- The tree parameters are shared across different tree levels; in other words, there are mere $K$ vectors (corresponding to the $K$ different branches) in each K-ary tree. Thanks to parameter sharing, RecForest becomes much more memory-efficient compared with the existing tree-based recommenders, as demonstrated in Table 1 theoretically and in Table 3 empirically.

- Given the above settings, RecForest becomes much less sensitive to the partition of the item set. As a result, without any tree update, RecForest can perform remarkably better than TDM and JTM with repetitive tree update, as shown in Table 3. Therefore, RecForest can avoid the repetitive adaption of the tree structure, saving a considerable portion of the training cost.

We perform comprehensive evaluations on six popular recommendation datasets. According to our experiment results, RecForest notably outperforms the existing tree-based recommenders in terms of recommendation quality and efficiency. Besides, we empirically verify that RecForest can be effectively trained with much less time cost, indicating its strong usability in real-world scenarios.

## 2 Recommender Forest

### 2.1 Preliminary

As mentioned above, RS usually represents user and item as latent embeddings, where the user's preferences for the items are measured by the inner product of the embeddings [19, 1, 11, 12]. As a result, the top-n recommendation is boiled down to the maximum inner product search (MIPS) problem. For the sake of efficient recommendation, various ANNs indexes are leveraged in practice, such as the tree-based index [18, 10, 2, 14]; the hash-based index [21, 22, 7, 17], the quantization-based index [4, 25, 5, 13, 15] and the graph-based index [16, 23, 27, 3], etc. These indexes have been well implemented by toolkits, like FAISS, which greatly facilitates the deployment of recommendation systems in practice. However, one limitation of the current ANN-based methods is that the index construction and the representation learning are decoupled, which will probably introduce incompatibility between both modules.

To mitigate the above problem, recent works were proposed to jointly optimize the representation model and index, where two representative works are TDM [29] and JTM [28]. Both methods are

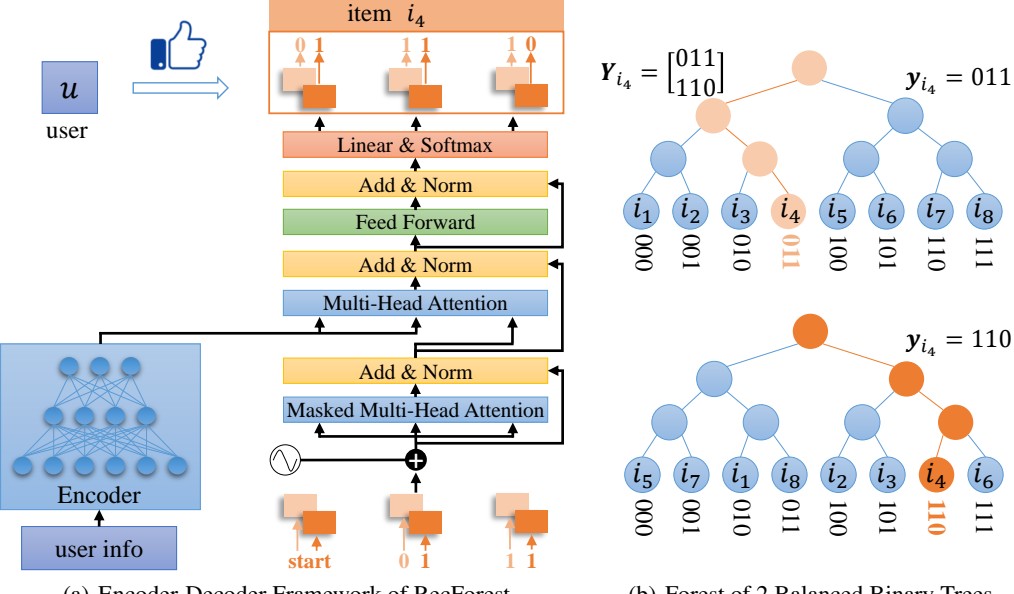

(a) Encoder-Decoder Framework of RecForest

(b) Forest of 2 Balanced Binary Trees

Figure 2: Illustration of RecForest framework.

based on the tree structures, where the top-n recommendation can be efficiently accomplished by a layer-wise beam search. For example, as shown in Figure 1, when the beam size is 2, these models will score the four children of previously retrieved 2 nodes, from which the two largest children will be selected. Such a process is iteratively performed until the final top-2 leaf nodes are retrieved.

## 2.2 Overview

The framework of RecForest is overviewed as follows. First of all, the item set is partitioned based on hierarchical balanced Kmeans, so that a tree structure is generated. The tree structure will enable an item to be efficiently retrieved in $O(\log N)$ time complexity ($N$ is the number of the items). Knowing that the items located close to the partition boundaries may get falsely assigned to a branch missing from the retrieval process, we propose to leverage multiple diversified trees, where the missing probability can be largely reduced. One item is corresponding to a leaf node in the tree, which can be represented by a path stretching from the route. Suppose that the K-ary tree is utilized, each item can be represented by a sequence of branch ID $(0, ..., K-1)$, whose length is $\lceil \log_K N \rceil$. For example, in Figure 2(b), the item $i_4$ is represented by "011" and "110" in the corresponding binary trees, respectively. Given the hierarchical

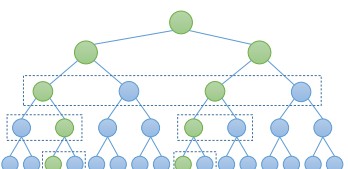

Figure 1: Beam search over a binary tree. The beam size is set 2. The nodes in the dotted box of each layer are candidates for scoring, where green nodes are selected.

numbering of the items, the recommendation turns out to be a sequence-to-sequence problem: based on the encoded user representation, paths to the most preferable items are progressively decoded via beam search, from which the top-n recommendation is made. The overall framework of RecForest is shown in Figure 2(a).

For ease of understanding, let's first summarize the notations used in this paper. Let $\boldsymbol{x}_u$ denote the user information of $u$, which can be either a behavior sequence or a user feature vector. Let $\boldsymbol{y}_i \in \{0, K-1\}^H$ be the item $i$'s routing sequence of length $H$ and $\boldsymbol{Y}_i \in \{0, K-1\}^{T \times H}$ be a concatenation of item $i$'s routing sequences of $T$ trees. Formally, RecForst models the conditional probability $P(\boldsymbol{Y}_i | \boldsymbol{x}_u)$ with the seq2seq architecture.

### 2.3 Construction of Tree and Forest

As mentioned, the item set is organized by a tree structure, where semantically close items should fall into the same branch as much as possible. We will construct multiple trees so as to alleviate the false assignment of items around partition boundaries. Detailed workflows are illustrated as follows.

#### 2.3.1 Tree Construction

The tree is expected to be balanced for the sake of efficient beam search on the tree. Thus, we leverage hierarchical balanced clustering, where each cluster is evenly divided into $K$ child clusters until each child cluster merely contains one single item. We propose to randomly sample $K^H - N$ items ($H = \lceil \log_K N \rceil$ is the height of the tree) from the entire item set, such that the constructed tree will be a complete K-ary tree.

We suggest the following two methods for the balanced partition of the item set. **Random**: For each cluster, the included items are randomly partitioned into $K$ equally-sized subsets. Such a method is extremely simple; yet, the semantic relationship between items is ignored. **Balanced Kmeans** (short for Kmeans): Particularly, we first pre-train item embeddings based on an arbitrary off-the-shelf recommendation model. In our experiment, we use the Deep Interest Network (DIN) [26] for sequential recommendation and the Bayesian Personalized Ranking (BPR) [19] for non-sequential recommendation, given their effectiveness and popularity in corresponding scenarios. With such a recommendation model, the items' semantic closeness can be measured by their embedding similarity, as relevant items will be close to each other in the latent space. Then, for each cluster, the included items are evenly partitioned via Kmeans w.r.t. their embedding similarity. Due to the space limitation, details about balanced Kmeans clustering are given in the appendix.

#### 2.3.2 Forest Construction

Different trees are desirable of being diversified such that the boundary items can be better covered from beam search. To this end, we propose a simple but effective approach, which combines the random and Kmeans tree construction. Particularly, for each cluster with $n$ items, $n \mod K$ items are first randomly chosen, from which one item is sampled with replacement into each child cluster; then, the remaining items are further partitioned into the K equally-sized groups via balanced Kmeans. By doing so, different trees are naturally diversified due to the inherent randomness.

### 2.4 RecForest Enocoder

As shown in Figure 2(a), the encoder is utilized to encode user information and output user representation. Any neural network based encoders are compatible with our framework. We mainly consider the sequential scenario and non-sequential scenario, where a sequence encoder and a feature encoder is used to encode user behavior sequences and feature vectors, respectively.

**Feature Encoder** In the non-sequential scenarios, user information is denoted by a feature vector, i.e. $\boldsymbol{x}_u$. We simply use a MLP for the feature encoder. $\boldsymbol{z}_u = \mathrm{MLP}(\boldsymbol{x}_u)$. When only user id is available, we can take its embedding as user representation, without any non-linear transformation.

**Sequence Encoder** In the sequential scenarios, user information is an item sequence. Concretely, $\boldsymbol{x}_u = [i_{t_1}, i_{t_2}, \ldots, i_{t_m}]$, where the item $i_{t_j} (1 \leq j \leq m)$ is the $j$-th interacted item of user $u$. It is straightforward to take item features into account. In view of Transformer's [24] state-of-the-art performance in seq2seq tasks like machine translation and speech recognition, we utilize the Transformer encoder for the sequence encoder. In particular, $\boldsymbol{z}_u = \mathrm{transformer\text{-}encoder}(\boldsymbol{x}_u)$, where the length of $\boldsymbol{z}_u$ is the same as $\boldsymbol{x}_u$. Due to space limitation, more details about the Transformer encoder are provided in the appendix.

### 2.5 RecForest Decoder

The decoder is to predict the next branch (routing decision) given the routing trajectory and the encoder's outputs of user representation. As shown in Figure 2(a), we apply the Transformer [24] decoder for this task due to its powerful capability. From down to up, following the branch embedding and positional embedding, the Masked Multi-Head Attention [24] is utilized to model the complex

dependence among branches at different layers. Taking the outputs from the Masked Multi-Head Attention, Multi-Head Attention [24] is followed to encode the complex interactions between user representation and routing trajectory representations. The FFN layer then enhances non-linearity of context-aware routing trajectory representations. A linear layer with the softmax activation is applied for predicting the next routing decision. Note that this decoder can also handle the non-sequence scenarios since a user feature vector can be considered as a sequence only with one element.

## 2.6 Training

As depicted in Figure 2(b), the routing trajectory of item $i_4$ is [0,1,1] and [1,1,0] on the two trees, respectively. The framework uses [start,0,1] to predict [0,1,1] and uses [start,1,1] to predict [1,1,0]. 'start' is a special symbol for initializing the sequence, which actually corresponds to the tree root. According to experimental results, we do not share the transformer decoder among trees, i.e., $P(\boldsymbol{Y}_i|\boldsymbol{x}_u) = \prod_k P_{\theta_k}(\boldsymbol{Y}_i[k]|\boldsymbol{x}_u)$, where $\theta_k$ indicate the parameters of the $k$-th tree. In this case, all trees share the same training procedure, so we illustrate the training with one tree. At each layer, the training task is to predict the next routing decision given routing trajectory, which corresponds to a multi-class classification problem. Given a user $u$ and its interacted item $i$, the loss for optimizing encoder and decoder is formulated as follows:

$$\mathcal{L}(u,i) = -\sum_{h=0}^{H-1} \log \operatorname{Prob}(y_i^h | \operatorname{cat}(\operatorname{start}, \boldsymbol{y}_i^{[0:h]}), \boldsymbol{z}_u) \tag{1}$$

where $y_i^h$ and $\boldsymbol{y}_i^{[0:h]}$ indicate the branch (routing decision) at the h-th layer and routing trajectory until the h-th layer respectively, and 'cat' means concatenation.

## 2.7 Inference

Since each leaf in the tree corresponds to an item, RecForest transforms top-n recommendation into the sequence prediction problem. In particular, we first apply the encoder for deriving user representation and leverage the decoder to generate the top-n routing sequences, i.e., paths from the root node to leaf nodes on each tree, based on beam search. Note that each item can correspond to multiple leaf nodes (i.e. routing sequences) in each tree since some items are randomly sampled for building a complete K-ary tree. Therefore, the generated items should be duplicated. Let $C$ denote the union of generated candidate items from the forest. For $c \in C$, we first compute the logarithmic probability $\sum_{h=0}^{H-1} \log \operatorname{Prob}(y_i^h | \operatorname{cat}(\operatorname{start}, \boldsymbol{y}_i^{[0:h]}), \boldsymbol{z}_u)$ in each tree and then sum logarithmic probabilities over all trees as the score of the item $c$. We then select the top-n items from $C$ for recommendation.

## 2.8 Complexity Analysis

**Notation**     Denote by $D$ item embedding size, $B$ the beam search size, $K$ the number of branches, and $I$ the training set size. Denote by $T$ the number of trees in RecForest and the times of updating the tree in JTM and TDM. In SCANN, denote by $K_{vq}$ the number of centroids in VQ, $K_{pq}$ the number of centroids in PQ, $M$ the number of subspaces and $W$ the number of probed VQ cells.

**Time Complexity**     In inference, the beam search can be finished within $O(TKB \log_K N)$ since the probabilities of $KB$ branches at each level of all trees need to be calculated. Hierarchical Kmeans clustering can be done within $O(NK \log_K N)$ to build a tree index so that it takes $O(TNK \log_K N)$ to build the forest index.

**Space Complexity**     The memory consumption of the recommendation model is inevitable in any recommender systems, so we mainly focus on the size of the index structure. In RecForest, we only need to keep the tree structure and $T$ matrices with size $K \times D$, denoting branch embeddings of $T$ trees. In TDM and JTM, beside the tree structure, they also store all tree node embeddings, where the number of tree nodes is at the same magnitude as the number of the items.

We summarize complexity analysis of typical algorithms in Table 1, which shows RecForest enjoys a small index memory cost and low inference time. This is because a smaller beamsize and fewer beam search are only required, and tree node representations are not stored but computed on the fly.

Table 1: Complexity analysis. The SCANN indexing time only considers the encoding period given codebooks for fair comparison. The time complexity of IPNSW directly is taken from the original paper of HNSW, which is derived with exact Delaunay graphs.

| Complexity | RecForest | JTM | TDM | IPNSW | SCANN |
|---|---|---|---|---|---|
| Inference Time | $O(TKB \log_K N)$ | $O(B \log N)$ | $O(B \log N)$ | $O(\log N)$ | $O(K_{vq} + K_{pq} + \frac{W}{K_{vq}} \cdot N)$ |
| Indexing Time | $O(TNK \log_K N)$ | $O(TI \log N)$ | $O(TN \log N)$ | $O(N \log N)$ | $O(NMK_{pq})$ |
| Index Size | $O(TKD)$ | $O(ND)$ | $O(ND)$ | $O(ND)$ | $O(K_{vq}D + K_{pq}D)$ |

# 3 Experiments

We conduct the experiments to answer the following research questions: **RQ-1**: *Does RecForest outperform the SOTA efficient recommenders in the tradeoff between efficiency and accuracy?* **RQ-2**: *Does the forest-based index improve near-boundary item retrieval?* **RQ-3**: *How much effect positional embedding can take?* The experiments are carried out in both sequential scenarios and non-sequential scenarios, where the sequence encoder and feature encoder is utilized, respectively. Due to space limitations, we only report the results in the sequential scenario. Other results are provided in the appendix. These experiments are done on a Linux server with Tesla V100 GPUs.

## 3.1 Dataset

We evaluate the RecForest with six real-world recommendation datasets, which can be downloaded from the url[*]. The datasets are **Movie-Lens 10M** (abbreviated as **Movie**), **Amazon Books** (abbreviated as **Amazon**), **Tmall Click** (abbreviated as **Tmall**), **Gowalla Check-in Dataset** (abbreviated as **Gowalla**), Microsoft News Dataset (abbreviated as **MIND**). Since some datasets only include rating-based explicit feedback, they should be converted into implicit feedback for RecForest's inputs. These datasets are pre-processed by filtering the users who interact with no more than 15 items. The overall information of datasets is summarized in Table 2.

Table 2: Statistics of Datasets

| Dataset | #User | #Item | #Interaction | Density |
|---|---|---|---|---|
| Movie | 69,878 | 10,677 | 10,000,054 | 1.34% |
| Amazon | 29,980 | 67,402 | 2,218,926 | 0.11% |
| Tmall | 139,234 | 135,293 | 10,487,585 | 0.05% |
| Gowalla | 13,583 | 71,436 | 977,425 | 0.10% |
| MIND | 36,281 | 7,129 | 5,610,960 | 2.16% |
| Yelp | 26,031 | 35,294 | 1,713,759 | 0.19% |

## 3.2 Baselines

In the part, we report the results of the sequential recommendation scenario, so we compare the proposed RecForest with two-stage indexes, IPNSW [16] and SCANN [5], with learnable indexes, TDM [29] and JTM [28] as well as with YoutubeDNN [1] and DIN [26] based on the brute-force retrieval. Note that brute-force-based YoutubeDNN and DIN are time-prohibitive in online services, but they can be considered as two strong baselines. IPNSW and SCANN are built over item embedding of a recommender. The recommender shares the same encoder as RecForest, but replaces the decoder with flat item embedding. It is then trained with the same loss as DIN. Both TDM and JTM make a routing decision simply based on current node without the consideration of routing trajectories. More details about baselines can be referred to in the appendix.

## 3.3 Experimental Settings

In each dataset, we randomly choose 10% users as validation users, 10% users as test users, and all the left users as training users. Following TDM and JTM, we use a slide window to split user-item interaction histories into slices of length 70 at most. For training users' data, the first 69 interactions are used for input context and the 70-th item is regarded as the ground truth of prediction. For data of both validation users and test users, we regard the first half as context and others as ground truth.

The latent dimensionality is set to 96 in all methods. Here, for a fair comparison of running time, we implement SCANN and IPNSW with PyTorch. To construct the quantization-based index, we follow

---

[*]https://drive.google.com/drive/folders/1ahiLmzU7cGRPXf5qGMqtAChte2eYp9gI

Table 3: Comparison with Baselines w.r.t NDCG@20 and NDCG@40, index memory cost (MB), and inference time (second). The bold fonts indicate the best performance.

| | NDCG@20 | NDCG@40 | Memory | Time | NDCG@20 | NDCG@40 | Memory | Time |
|---|---|---|---|---|---|---|---|---|
| Method | **Movie** | | | | **Amazon** | | | |
| DIN | **0.5440** | 0.5473 | - | 193.87 | **0.2766** | **0.3039** | - | 492.64 |
| YoutubeDNN | 0.5329 | **0.5484** | - | **29.38** | 0.2195 | 0.2491 | - | **120.91** |
| JTM | 0.5149 | 0.5075 | 10.80 | 12.05 | 0.1533 | 0.1683 | 75.99 | 6.64 |
| TDM | 0.4684 | 0.4651 | 10.80 | 9.33 | 0.0856 | 0.0949 | 75.99 | 6.61 |
| SCANN | 0.4665 | 0.4695 | 3.64 | 18.64 | 0.1529 | 0.1780 | 14.66 | 4.48 |
| IPNSW | 0.5330 | 0.5486 | 10.08 | 15.52 | 0.2255 | 0.2548 | 66.46 | 10.28 |
| RecForest | **0.5580** | **0.5682** | **3.21** | **8.33** | **0.2339** | **0.2576** | **7.32** | **3.79** |
| Method | **Gowalla** | | | | **Tmall** | | | |
| DIN | **0.2798** | **0.3095** | - | 186.41 | **0.2275** | **0.2491** | - | 4057.69 |
| YoutubeDNN | 0.2312 | 0.2637 | - | **53.55** | 0.1736 | 0.1975 | - | **1086.75** |
| JTM | 0.2595 | 0.2484 | 77.56 | 2.64 | 0.0749 | 0.0849 | 151.19 | 30.11 |
| TDM | 0.1723 | 0.1775 | 77.56 | 2.55 | 0.0257 | 0.0272 | 151.19 | 29.42 |
| SCANN | 0.1839 | 0.2083 | 15.48 | 1.86 | 0.1105 | 0.1226 | 28.10 | 20.88 |
| IPNSW | 0.2464 | 0.2805 | 70.39 | 4.73 | 0.1696 | 0.1902 | 132.72 | 52.90 |
| RecForest | **0.3783** | **0.3963** | **7.39** | **1.82** | **0.2059** | **0.2261** | **9.29** | **18.88** |
| Method | **MIND** | | | | **Yelp** | | | |
| DIN | **0.7399** | **0.7399** | - | 62.98 | **0.2825** | **0.3117** | - | 170.25 |
| YoutubeDNN | 0.7349 | 0.7336 | - | **52.14** | 0.2518 | 0.2850 | - | **48.51** |
| JTM | 0.5956 | 0.5505 | 6.62 | 5.48 | 0.1014 | 0.1300 | 39.47 | 4.21 |
| TDM | 0.5615 | 0.5198 | 6.62 | 5.51 | 0.1547 | 0.1515 | 39.47 | 4.34 |
| SCANN | 0.5987 | 0.5713 | 3.20 | 19.51 | 0.1729 | 0.2012 | 8.44 | 3.58 |
| IPNSW | 0.7346 | 0.7331 | 6.95 | 8.99 | 0.2562 | 0.2906 | 34.74 | 8.92 |
| RecForest | **0.7583** | **0.7579** | **3.18** | **4.61** | **0.2766** | **0.3031** | **6.81** | **3.57** |

SCANN's default settings. To construct the graph index of IPNSW, the maximum degree of each node is set to 16. The beam size for any beam search is set to 100 unless specified. To learn models, the learning rate is all set to 1e-3 with exponential decay. The items' representations for constructing trees are obtained from item embedding of the pre-trained DIN on each dataset.

## 3.4 Comparison with Baselines

**Settings**  We compare RecForest with the baselines on all datasets. RecForest uses 2 trees on the Movie and MIND, and 4 trees on other datasets. Other settings can be referred to in Section 3.3.

**Results**  All the results are shown in Table 3, where the index memory cost of RecForest indicates the decoder's memory consumption. The following observations answers the **RQ-1**.

- **RecForest significantly outperforms all efficient recommenders with indexes on all datasets w.r.t NDCG@20 and NDCG@40**. The improvements over the best baselines w.r.t NDCG@20 are 4.69%, 3.72%, 45.78%, 21.40%, 3.23%, 7.96% on the Movie, Amazon, Gowalla, Tmall, MIND, and Yelp dataset, respectively. Its superiority over two-stage baselines (i.e., IPNSW and SCANN) demonstrates the effectiveness of joint representation learning, although the graph-based index shows a strong performance. The higher accuracy than TDM and JTM indicates the benefit from the use of routing trajectory, multiple trees and the powerful transformer decoder.

- **RecForest performs better than brute-force-based YoutubeDNN on all datasets and even than brute-force-based DIN on the Movie, Gowalla, and MIND datasets.** Note that we do not apply DIN for post-reranking, otherwise the retrieval accuracy can be further improved. This again confirms the superpower of the transformer decoder. Note that after fine-tuning the hyper-parameters, YoutubeDNN performs well compared to TDM and JTM, which may not be consistent with the results in TDM and JTM.

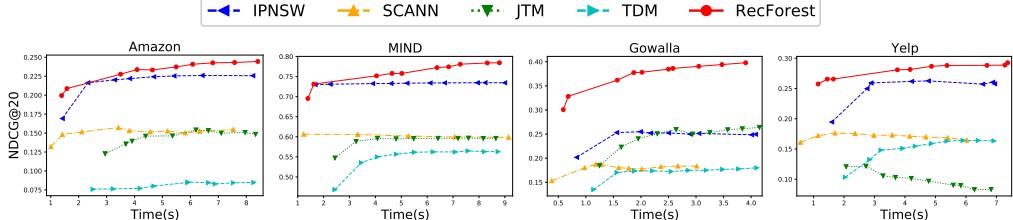

Figure 3: Tradeoff between Efficiency and Accuracy

- **RecForest also enjoys the smallest index memory cost compared to other methods**, which is now supported empirically and theoretically. This is because RecForest only embeds the routing decision and compute tree node representation on the fly, compared to directly embedding all tree nodes in TDM and JTM. RecForest even costs fewer memories for index than the most lightweight PQ-based index – SCANN, which only stores the codes of data points and codebooks. In spite of good retrieval accuracy, the graph-based index IPNSW is memory-heavy due to the storage of graphs and data points. Note that both DIN and YoutubeDNN are based on exhaustive search so they don't consume memory for indexes.

- **RecForest is almost as fast as and sometimes even faster than SCANN for the top-n item retrieval**, which is considered as the SOTA index for MIPS in retrieval efficiency and accuracy. This benefits from transforming the top-n recommendation into sequence prediction, which can be efficiently accomplished by beam search on GPU. Note that the running time cost of IPNSW seems inconsistent with Table 1, this is because only the approximate proximity graphs instead of exact Delaunay graphs can be efficiently constructed for query in practice. Theoretical results show that the RecForest is almost as efficient as TDM and JTM, while the experimental results indicate RecForest is slightly faster. On the one hand, this is because we use the comparatively large branches (much larger than 2 used in TDM and JTM), remarkably decreasing the tree depth and thus reducing the times of beam search. On the other hand, because RecForest uses the simple function for routing decision and enables high parallelism due to the use of transformer decoder.

### 3.5 Extensive Study between Efficiency and Accuracy

**Settings** The efficiency-accuracy curve is a commonly-used standard for evaluating the ANNs index, so we also provide this result for better illustrating the superiority of RecForest. The study is mainly investigated on the Amazon, MIND, Gowalla, and Yelp dataset, since the other two datasets show a similar trend. To vary the retrieval time of RecForest, we adjust the beam size of beam search from 10 to 100 with step 10, where the forest consists of at most 5 trees. For TDM, JTM, and IPNSW, we adjust the beam size from 10 to 200 with step 10. For SCANN, we adjust the number of probe cells from 50 to 2,000 with step 100.

**Results** The curves of different algorithms between NDCG@20 and query time are shown in Figure 3. The following observations confirms the answer of **RQ-1**. First, **RecForest strikes the best balance between query time and retrieval accuracy on all the four datasets**, since the curve stands on top of the others. This confirms the superiority of RecForest to these competing baselines. The advantage is the most significant on the Gowalla dataset, evidenced by the biggest gap between RecForest and IPNSW. Second, **With the increase of beam size, the accuracy of RecForest can improve more significantly than baselines.** This benefits from the powerful transformer decoder and the novel training paradigm as well as the forest-based index, such that the representation model and tree indexes are jointly learned better.

### 3.6 Ablation Study

#### 3.6.1 Effect of Forest Construction

**Settings** We investigate three aforementioned ways (in Section 2.3.2) of constructing a forest with at most 10 trees, where the branch number of each tree is set to 4.

**Results** The results on the Amazon, Gowalla, MIND, and Yelp datasets are reported in Figure 4. We can observe that **Kmeans with randomness (i.e. Random+Kmeans) always performs best and improves with the increasing number of trees while the Random performs worst and does not improve as much as Random+Kmeans when the tree number increases.** The reason why the Random does not work well is that the semantic requirement is not satisfied. Since Kmeans utilizes the semantic information, it performs better than the Random, but is not affected a lot by growing the number of trees. This can be explained by the lack of diversity between trees. This answers the **RQ-2**: since the use of forest-based index can lead to significant improvement of retrieval accuracy, particularly when tree construction takes semantics into account, *the forest-based index improves near-boundary item retrieval.*

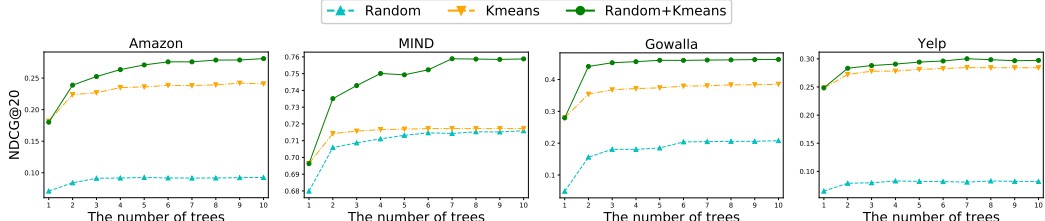

Figure 4: Ablation study: Different construction ways for forests.

### 3.6.2 Effect of Branch Number K

**Settings** We conduct evaluations on a single Kmeans-constructed tree on the Amazon, MIND, Gowalla and Yelp dataset, by varying the branch number in {2,4,8,10,16,18}.

**Results** The results are shown in Table 4. We can observe that **the retrieval accuracy of RecForest improves with the growing number of branches, particularly from 2 branches to 4.** The improvement on the Gowalla and Yelp datasets is the most significant. This observation can be explained as follows: First, the vocabulary size grows, so the decoder can be supervised by more challenging discriminative signals and the decoders' parameters grow. Secondly, the routing sequences would be shorted, reducing the error accumulation of sequence prediction to some extent.

Table 4: Effect of branch number

| #Branch | NDCG@20 | | | |
|---|---|---|---|---|
| | Amazon | MIND | Gowalla | Yelp |
| 2 | 0.1624 | 0.6900 | 0.2353 | 0.1951 |
| 4 | 0.1912 | 0.6935 | 0.2795 | 0.2269 |
| 8 | 0.1864 | 0.6966 | 0.2938 | 0.2483 |
| 10 | 0.2022 | 0.6954 | 0.2843 | 0.2577 |
| 16 | 0.2031 | 0.6977 | 0.2957 | 0.2437 |
| 18 | **0.2088** | **0.7154** | **0.3337** | **0.2581** |

### 3.6.3 Effect of Positional Embedding

**Settings** Since the routing sequence is used for representing an item, the order of routing decisions is important. To this end, we compare four positional embeddings: (1) **None** indicates that there is no positional embedding; (2) Absolute positional embedding (**Abs** for short) encodes the absolute positions which range from 1 to maximum length of sequence by learnable parameters; (3) Relative Key [20] (**RelK** for short) focuses on attention and relative distance between decisions; (4) Relative Key Query [8] (**RelKQ** for short) is a refined Relative Key. Please refer to [8] for more details.

**Results** The results are summarized in Table 5. We can observe that **positional embedding takes a remarkable effect on improving the retrieval capacity while different positional embedding methods do not have a significant difference in retrieval accuracy**, which answers the **RQ-3**. The effects of positional embedding are more significant on the Gowalla and Yelp datasets.

Table 5: Effect of positional embedding

| Pos. Emb. | NDCG@20 | | | |
|---|---|---|---|---|
| | Amazon | MIND | Gowalla | Yelp |
| None | 0.1704 | 0.6476 | 0.1697 | 0.1988 |
| Abs | 0.1805 | **0.6986** | **0.2632** | **0.2505** |
| RelK | 0.1824 | 0.6942 | 0.2617 | 0.2402 |
| RelKQ | **0.1840** | 0.6917 | 0.2588 | 0.2492 |

## 4 Conclusion and Future Work

In this paper, we propose the Recommender Forest for efficient recommendation, which can be simply trained within the sequence-to-sequence framework. RecForest enjoys a small index memory cost, low inference time, and highly-accurate recommendation even without updating the tree structure. The extensive study on six real-world recommendation datasets shows that RecForest becomes a state-of-the-art efficient recommender. In the future, we explore multi-task learning, non-autoregressive prediction, and index structure learning as well as a more general framework.

## Acknowledgement

The work was supported by grants from the National Natural Science Foundation of China (No. 62022077 and 61976198)

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
