# Appendix

# A    Forest Construction

## A.1    Balanced Kmeans

For the sake of retrieval efficiency, the constructed tree is expected to be a complete K-ary tree. However, the item number $N$ does not necessarily meet this requirement. We grow the dataset by inserting $K^H - N$ randomly sampled items from itself, such that it exactly contains $K^H$ items, where $H = \lceil \log_K N \rceil$ is the tree height. We use the Balanced Kmeans in Algorithm 1 for hierarchical itemset partition. In particular, supposing a cluster contains $K \times m$ items, the algorithm splits them into $K$ clusters such that each cluster exactly contains $m$ items. At each iteration, we first assign each centroid the $m$ closest unassigned items in order and then update the centroids according to item assignment. The clustering algorithm terminates until the assignments do not change any more.

---

**Algorithm 1:** Balanced-Kmeans

    **Data:** Item set $\mathbb{D}$, cluster number $K$
    **Result:** $\{\mathbb{D}_1, \mathbb{D}_2, \ldots, \mathbb{D}_K\}$
**1** $m \leftarrow \frac{|\mathbb{D}|}{K}$;
**2** Randomly initialize $K$ centroids $\mathbb{C} = \{\boldsymbol{c}_1, \boldsymbol{c}_2, \ldots, \boldsymbol{c}_K\}$ ;
**3** **repeat**
**4**      $\mathbb{U} \leftarrow \mathbb{D}$ ;
**5**      **for** $k = 1 \rightarrow K$ **do**
**6**          $U_s \leftarrow$ Sort items in $\mathbb{U}$ ascendingly according to distance from $\boldsymbol{c}_k$;
**7**          $\mathbb{D}_k \leftarrow U_s[0 : m - 1]$;
**8**          $\boldsymbol{c}_k \leftarrow \frac{1}{m} \sum_{\boldsymbol{x} \in \mathbb{D}_k} \boldsymbol{x}$;
**9**          $\mathbb{U} \leftarrow \mathbb{U} \setminus \mathbb{D}_k$;
**10** **until** *Assignments do not change*;

---

## A.2    Tree Construction

As described in Section 2.3.2, each tree is constructed by combining **Random** partition and Balanced **Kmeans** to guarantee both diversity and semantic closeness of similar items. Regarding the **Random** strategy, for each cluster, the included items are randomly partitioned into $K$ equally-sized subsets. When it is combined with balanced Kmeans, for each cluster with $n$ items, $n \mod K$ items are first randomly picked, from which one item is sampled with replacement into each child cluster, and the remaining items are partitioned based on the Balanced Kmeans. This method is shorten for **Random+Kmeans**. By applying the **Random+Kmeans** strategy recursively until each cluster only contains one item, the tree index can be constructed successfully. Repeating the tree construction can generate diverse trees to form the forest. We summarize the procedure in Algorithm 2.

---

**Algorithm 2:** Tree construction

    **Data:** Dataset $\mathbb{D}$, branch number $K$, random ratio $\delta$
    **Result:** Constructed tree
**1** $Q \leftarrow$ A queue with an element $\mathbb{D}$;
**2** **while** $Q$ *is not empty* **do**
**3**      $\mathbb{D}_{top} \leftarrow \text{pop}(Q)$;
**4**      $n \leftarrow |\mathbb{D}_{top}|$;
**5**      $\mathbb{D}^0 \leftarrow$ Randomly draw $n \mod K$ items from $\mathbb{D}_{top}$ without replacement ;
**6**      $\mathbb{D}_{top} \leftarrow \mathbb{D}_{top} \setminus \mathbb{D}^0$;
**7**      $\{\mathbb{D}_1, \mathbb{D}_2, \ldots, \mathbb{D}_K\} \leftarrow$ Balanced-Kmeans$(\mathbb{D}_{top}, K)$;
**8**      **for** $k = 1 \rightarrow K$ **do**
**9**          $i \leftarrow$ Sample one item with replacement from $\mathbb{D}^0$;
**10**          push$(Q, \{i\} \cup \mathbb{D}_k)$;

---

# B  Transformer encoder

The Transformer encoder, shown in Figure 5, is to capture user's interests from the sequential interactions $\boldsymbol{x}_u = [i_{t_1}, i_{t_2}, \ldots, i_{t_m}]$. From down to up, following the interaction embedding and positional embedding, Multi-Head Attention [24] is utilized to model the interdependence of user's each behavior. Taking the output of Multi-Head Attention, the FFN layer is to enhance the non-linearity of user's behavior representations. Finally, the encoder output, denoted as $\boldsymbol{z}_u = [\boldsymbol{z}_u^1; \boldsymbol{z}_u^2; \ldots; \boldsymbol{z}_u^m]$, is utilized to characterize user representation.

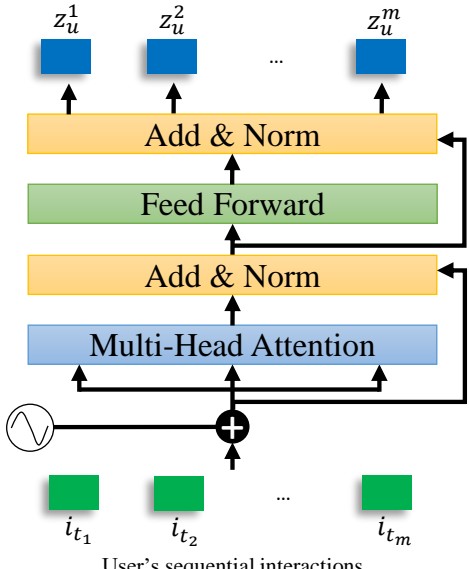

Figure 5: Transformer Encoder

# C  Pseudocode of training and inference

In our RecForest, all the trees share the encoder but each tree has a unique decoder. The training framework is summarized in Algorithm 3. Note that we only give one training instance for clear illustration and it can be done in batch-wise manner for practical implementation. In addition, the inference framework is summarized in Algorithm 4. Once the generated candidates from each tree are collected, we compute the generated probabilities of candidates on each tree for ranking purpose.

---

**Algorithm 3:** RecForest Training

    **Data:** $Forest=\{tree_1, tree_2, \ldots, tree_T\}$, training instance $(x_u, i)$, Encoder and Decoder$_t$
        $(1 \leq t \leq T)$ w.r.t. each tree

1   $\boldsymbol{z}_u \leftarrow$ Encoder($\boldsymbol{x}_u$);
2   **foreach** $t \in \{1, 2, \ldots, T\}$ **do**
3      target$\leftarrow$ get the path of item $i$ on the $tree_t$;
4      decoder-input$\leftarrow$ ['start',target[0:H-1]]; // H is tree height
5      prediction$\leftarrow$ Decoder$_t$($\boldsymbol{z}_u$,decoder-input);
6      $\mathcal{L}(u, i) \leftarrow$ loss(prediction, target[1:H]) according to Eq (1);
7      back-propagation from $\mathcal{L}(u, i)$;
8      update Encoder and Decoder$_t$ by gradient descent;

---

**Algorithm 4:** RecForest Inference

---

**Data:** $Forest=\{tree_1, tree_2, \ldots, tree_T\}$, Encoder and $Decoder_t$ ($1 \le t \le T$) w.r.t. each tree, user feature $x_u$, beam size $B$, $n$ (i.e. number of returning items)

**Result:** The top-$n$ items for user $u$

1  $R \leftarrow \{\}$;
2  $z_u \leftarrow \text{Encoder}(x_u)$;
3  **for** $t \in \{1, 2, \ldots, T\}$ **do**
4       paths $\leftarrow$ Beam-Search($Decoder_t, z_u, B$) ;
5       **foreach** $p \in paths$ **do**
6           item $i \leftarrow$ get-item($p, tree_t$);
7           $R \leftarrow R \cup \{i\}$;

8  **foreach** *item* $i \in R$ **do**
9       $score[i] = 0$;
10      **for** $t \in \{1, 2, \ldots, T\}$ **do**
11          $path \leftarrow$ get-path($i, tree_t$);
12          $p_t(i) \leftarrow$ compute the $path$ probability based on $Decoder_t$;
13          $score[i] = score[i] + \log p_t(i)$;

14 select the top-$n$ items from $R$ based on $score$;

---

## D  Metrics

The ranking metric, Normalized Discounted Cumulative Gain($NDCG$), is used to measure the performance of recommendation engine. Discounted Cumulative Gain($DCG$) can be calculated by

$$DCG@k = \sum_{i=1}^{k} \frac{2^{r_i} - 1}{\log_2(i+1)}$$

where $r_i$ means the relevance score and $k$ is the number of returned items. As we focus on implicit feedback, $r_i = 1$ if the test user really interacts with the returned $i$-th item otherwise $r_i = 0$. NDCG can be calculated by

$$NDCG@k = DCG@k/iDCG@k$$

where $iDCG$ means ideal $DCG$, i.e. ranking all the returned items according to relevance scores in a descending order and calculating the $DCG$ on the ordered items. Additionally, we also concern the memory consumption of retrieval index and inference efficiency. Concretely, we record the model size of retrieval system and the time consumption in inference.

## E  Experiment Details in sequential recommendation scenario

### E.1  Baselines in sequential recommendation scenario

- **TDM** [29] and its variant **JTM** [28], proposed by Alibaba Group, are the tree-based recommender frameworks which train the user-item preference model and the tree structure simultaneously. The leaf nodes represent items and the internal nodes represent the clusters which the corresponding leaf nodes belong to. Beam search is conducted on the tree to retrieve the items from coarse-grain clusters to fine-grain clusters. The main difference between TDM and JTM is the updating of bijective mapping relation between items and leaf nodes.

- Deep Interest Network (i.e. **DIN**) [26] utilizes the deep model to characterize the user's preferences over items from user's sequential interactions. Concretely, DIN casts the training task as a binary-class classification that reflects whether the user interacted with certain item. As DIN can't be used for efficient recommendation, we report the brute-force-based performance.

- **YoutubeDNN** [1], proposed by Google, is a deep neural network model which learns the user representation and item representation from sequential user-item interaction history.Then the inner product between user representation and item representation is used to capture user's preference over the corresponding item, which converts the recommendation task to maximum inner product

search(MIPS). In our experiments, we mainly show the performance of YoutubeDNN by burte-force search for MIPS.

- To solve MIPS efficiently, Quantizaiton-based method **SCANN** [5] and graph-based method **IPNSW** [16] meet the start-of-the-art performance in each technique route respectively. For fair comparison with our model, we model user's interactions by Transformer encoder and replace the decoder by item embedding. The adapted model is trained in the same way as DIN to obtain the user representation and item representation so that the learned representation can be used to build the SCANN index and IPNSW index.

The open-source code of TDM[*], JTM[*] and DIN[*] are based on Alibaba Group's X-DeepLeraning framework, we re-implement them by Pytorch. And the open-source code of YoutubeDNN[*] is also available.

### E.2 Detailed settings in sequential recommendation scenario

In each dataset, we randomly choose 10% users as validation users, 10% users as test users and all the left users as training users. Following the settings in TDM [29] and JTM [28], we use a slide window to cut user-item interaction histories into slices of length at most 70. Each window contains at least 15 interactions and zero padding is applied if there is less than 70 interactions in the window. At the training stage, the first 69 user-item interactions are regarded as context and the 70-th item is regarded as ground truth in each window. For validation users and test users, we regard the second half interactions as ground truth and the first half interactions as context. The hidden size is set to 96 for all methods.

The Deep Interest Network (DIN) [26], which contains the embedding layer followed by a MLP with size [128,64,2] and outputs the like/dislike probability, is used as the the preference model for TDM and JTM. YoutubeDNN [1] contains embedding layer followed by a MLP with size [128,64,96] and the number of negative sampling is set to 1,000. For SCANN, we follow the default settings of the open sources[*] but implement them with Python and Pytorch for fair efficiency comparison. Concretely, the number of leaf is 2,000; the number of sub-space is 24 and the dimensionality of each sub-space is 4; the number of codewords in each sub-space is set to 16; the value of threshold is set to 0.2. To construct the graph index of IPNSW, the maximum degree of each node is set to 16, and the beam size is set to 100.

For our RecForest, YoutubeDNN, TDM, JTM and DIN, Adam [9] is used as the optimizer and learning rate is set to 1.0e-3 with exponential decay. In inference, the beamsize of RecForest, TDM, JTM and IPNSW is set to 100 unless specification. For RecForest, the branch number is set to 18 and the tree number is set to be 2 on Movie and MIND, and 4 on the left datasets unless specification. The items' representations which are utilized to construct the forest for RecForest are obtained by pre-train the DIN model on each dataset.

## F  Experiments in non-sequential scenarios

In non-sequential scenarios, we only apply user id to denote a user and the feature encoder to model user representation. Specifically, the learnable embedding vector of the user id denotes user representation.

### F.1  Baselines

The classical deep model, Neural Collaborative Filtering (**NCF**) [6], characterizes user's preference over item given the user id and item id. Due to the compatibility with any advanced user-item (or user-node) preference model, we utilize NCF for preference modeling in **TDM** [29] and **JTM** [28] frameworks. For **YoutubeDNN**, we replace the user representation learning part used in sequential

---

[*]https://github.com/alibaba/x-deeplearning/tree/master/xdl-algorithm-solution/TDM
[*]https://proceedings.neurips.cc/paper/2019/hash/1c6a0198177bfcc9bd93f6aab94aad3c-Abstract.html
[*]https://github.com/alibaba/x-deeplearning/tree/master/xdl-algorithm-solution/DIN
[*]https://github.com/shenweichen/DeepMatch
[*]https://github.com/google-research/google-research/tree/master/scann

Table 6: Comparison with Baselines w.r.t NDCG@20 and NDCG@40, index memory space (MB), and inference time (second). The bold fonts indicate the best performance.

| | NDCG@20 | NDCG@40 | Memory | Time | NDCG@20 | NDCG@40 | Memory | Time |
|---|---|---|---|---|---|---|---|---|
| Method | **Movie** | | | | **Amazon** | | | |
| NCF | 0.1679 | 0.2252 | - | 52.61 | 0.0433 | 0.0703 | - | 447.30 |
| YoutubeDNN | 0.2002 | 0.2532 | - | 39.98 | 0.1004 | 0.1284 | - | 129.57 |
| JTM | 0.2142 | 0.2535 | 10.80 | 1.62 | 0.0544 | 0.0799 | 75.99 | 2.57 |
| TDM | 0.2164 | 0.2536 | 10.80 | 1.62 | 0.0251 | 0.0322 | 75.99 | 2.43 |
| SCANN | 0.2227 | **0.2646** | 3.64 | 1.93 | 0.0936 | 0.1168 | 14.66 | 2.56 |
| IPNSW | 0.2014 | 0.2014 | 10.08 | 3.64 | 0.1022 | 0.1124 | 66.46 | 2.80 |
| RecForest | **0.2229** | 0.2640 | **3.21** | **1.50** | **0.1064** | **0.1338** | 7.32 | **2.20** |
| Method | **Gowalla** | | | | **Tmall** | | | |
| NCF | 0.1106 | 0.1545 | - | 304.79 | 0.0340 | 0.0490 | - | 3039.39 |
| YoutubeDNN | 0.3494 | 0.3613 | - | 52.97 | 0.0820 | 0.1005 | - | 1158.76 |
| JTM | 0.1404 | 0.1512 | 77.56 | 2.00 | 0.0722 | 0.0880 | 151.19 | 4.49 |
| TDM | 0.1462 | 0.1562 | 77.56 | 2.09 | 0.0596 | 0.0736 | 151.19 | 5.36 |
| SCANN | 0.2078 | 0.2179 | 15.48 | 1.96 | 0.0650 | 0.0754 | 28.10 | 5.45 |
| IPNSW | 0.3472 | 0.3589 | 70.39 | 1.98 | 0.0671 | 0.0707 | 132.72 | 5.20 |
| RecForest | **0.3500** | **0.3651** | **7.39** | **1.91** | **0.0743** | **0.0905** | 9.29 | **3.85** |
| Method | **MIND** | | | | **Yelp** | | | |
| NCF | 0.4070 | 0.4392 | - | 21.20 | 0.1078 | 0.1471 | - | 93.59 |
| YoutubeDNN | 0.4194 | 0.4428 | - | 19.31 | 0.1633 | 0.1944 | - | 55.82 |
| JTM | 0.3476 | 0.3664 | 6.62 | 3.34 | 0.1262 | 0.1636 | 39.47 | 3.36 |
| TDM | 0.3655 | 0.3799 | 6.62 | 3.25 | 0.0472 | 0.0583 | 39.47 | 3.42 |
| SCANN | 0.3799 | 0.3822 | 3.20 | 3.45 | 0.1250 | 0.1523 | 8.44 | 4.10 |
| IPNSW | 0.4178 | 0.4399 | 6.95 | 3.85 | 0.1641 | 0.1942 | 34.74 | 3.59 |
| RecForest | **0.4308** | **0.4587** | **3.18** | 3.19 | **0.1720** | **0.2016** | **6.81** | **3.24** |

recommendation scenarios with the embedding vector of user id. By this way, YoutubeDNN can be compatible with non-sequential recommendation task. **SCANN** and **IPNSW** are built on the learned representation of YoutubeDNN. Note that brute-force-based YoutubeDNN and NCF are time-prohibitive in online services, but they can be considered as two strong baselines.

### F.2 Experimental settings

In each dataset, we randomly choose 10% users as validation users, 10% users as test users and all the left users as training users. For validation users and test users, we randomly select a quarter of the interacted items as ground truth, and put the rest into the training set. The fusion NCF [6], which contains the GMF and MLP where the neuron numbers are set to [512,256,128, 96] in each layer, is included. Other necessary settings are as described in Section E.2.

### F.3 Comparison with Baselines

**Settings**   We compare RecForest with the baselines on all datasets. RecForest uses 2 trees on the Movie and MIND, and 4 trees on other datasets. Other settings can be referred to in Section F.2.

**Results**   All the results are shown in Table 6, where the index memory cost of RecForest indicates the decoder's memory consumption. From the table, we have the following findings.

- **Recforest nearly outperforms all efficient recommenders with indexes on all dataset w.r.t. NDCG@20 and NDCG@40** except that SCANN outperforms RecForest slightly on Movie w.r.t NDCG@40. Its superiority over two-stage baselines is not so significant indicates that inner product may be capable enough for representation learning to some extent in this simple

recommendation scenario. The higher accuracy than TDM and JTM also verify the benefit from the use of routing trajectory, multiple trees and the powerful transformer decoder.

- **RecForest performs better than brute-force-based NCF and YoutubeDNN on nearly all datasets** except the results on Tmall where YoutubeDNN outperforms RecForest slightly. This phenomenon confirms the the superpower of transformer decoder.

- Being similar to the results of sequential recommendation scenarios, **RecForest has the advantage in index memory cost**, which can be highlighted by comparisons with baselines. These observations can be explained by referring to the same experiments in sequential scenarios. Note that NCF and YoutubeDNN are based on exhaustive search so they don't consume memory for indexes.

### F.4  Extensive Study between Efficiency and Accuracy

**Settings**    Similar to settings in sequential recommendation scenarios, we also draw the efficiency-accuracy curve. The study is mainly investigated on the Amazon, MIND, Gowalla, and Yelp dataset, since the other two datasets show a similar trend. As described in the sequential scenario, we adjust the beams size to vary the retrieval time cost for TDM, JTM, IPNSW and RecForest. For SCANN, we adjust the number of probe cells to control the time cost.

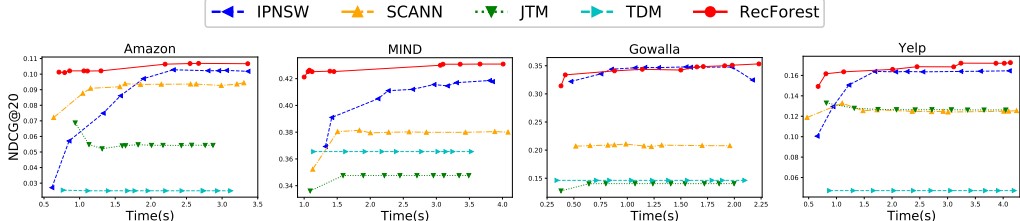

Figure 6: Tradeoff between Efficiency and Accuracy

**Results**    The curves of different algorithms between NDCG@20 and query time are shown in Figure 6. From the figure, we have the following findings.

- **RecForest strikes the best balance between query time and retrieval accuracy on all the four datasets**, since the curve stands on top of the others. This confirms the superiority of RecForest to these competing baselines. The advantage is the most significant on the MIND, evidenced by the biggest gap between RecForest and IPNSW.

- **With the increase of beam size, the accuracy of RecForest can improve more significantly than baselines.** This benefits from the powerful transformer decoder and the novel training paradigm as well as the forest-based index, such that the representation model and tree indexes are jointly learned better.

### F.5  Ablation Study

#### F.5.1  Effect of Forest Construction

**Settings**    We investigate three aforementioned ways of constructing a forest with at most 10 trees, where the branch number of each tree is set to 4. The details about forest construction can be referred to in Section 2.3.2.

**Results**    The results on the Amazon, MIND, Gowalla and Yelp datasets are reported in Figure 7. The results on Amazon, MIND and Yelp show the same trends as ones in sequential scenarios. Concretely, **Kmeans with randomness (i.e. Random+Kmeans) always performs best and improves with the increasing number of trees while the Random nearly performs worst and does not improve as much as Random+Kmeans when the tree number increases.** These results verify the findings again, i.e. the semantic information and the diversity among trees are important to improve the recommendation accuracy.

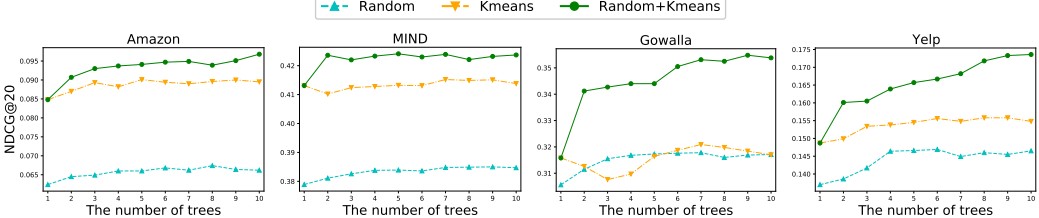

Figure 7: Ablation study: Different construction ways for forests.

### F.5.2 Effect of Branch Number K

**Settings**   We conduct evaluations on a single Kmeans-constructed tree on the Amazon, MIND, Gowalla and Yelp dataset, where the branch number varies in {2,4,8,10,16,18}.

**Results**   The results are shown in Table 7. We can observe that **the retrieval accuracy of RecForest on Amazon improves with the growing number of branches constantly but does not show a clear trend on other three datasets.** These findings indicate that the sensitivity of branch number depends on the dataset in non-sequential recommendation scenarios. Generally, the increase of branch numbers and the decrease of path length don't affect the performance as much as that in the sequential scenario.

Table 7: Effect of branch number

| #Branch | NDCG@20 | | | |
| --- | --- | --- | --- | --- |
| | Amazon | MIND | Gowalla | Yelp |
| 2 | 0.0724 | 0.4028 | 0.2916 | 0.1538 |
| 4 | 0.0869 | 0.4064 | 0.2837 | 0.1502 |
| 8 | 0.0886 | 0.3964 | 0.2947 | 0.1531 |
| 10 | 0.0913 | **0.4281** | 0.2949 | 0.1520 |
| 16 | 0.0961 | 0.4005 | **0.3052** | **0.1562** |
| 18 | **0.0971** | 0.4107 | 0.2978 | 0.1555 |

### F.5.3 Effect of Positional Embedding

**Settings**   We also investigate the effect of four positional embedding methods like what we do in the sequential recommendation scenario. Concretely, (1)**None**;(2)**Abs**;(3)**RelK**;(4)**RelKQ** and more details are referred to in Section 3.6.3.

**Results**   The results based on a single Kmeans-constructed tree on the Amazon, MIND, Gowalla, and Yelp datasets are summarized in Table 8. It is similarly observed to Section 3.6.3 that **positional embedding takes a remarkable effect on improving the retrieval capability while different positional embedding methods do not have a significant difference in retrieval accuracy**. The effect of Abs and RelK is more significant on Gowalla.

Table 8: Effect of positional embedding

| Pos. Emb. | NDCG@20 | | | |
| --- | --- | --- | --- | --- |
| | Amazon | MIND | Gowalla | Yelp |
| None | 0.0629 | 0.3656 | 0.2028 | 0.1140 |
| Abs | **0.0869** | **0.4064** | 0.2837 | 0.1502 |
| RelK | 0.0719 | 0.3975 | **0.2878** | **0.1509** |
| RelKQ | 0.0653 | 0.4049 | 0.2436 | 0.1443 |

## G   Additional Ablation Studies

To further verify the effect of Transformer Decoder and the joint training between the representation model and the ANN index, we conduct additional ablation studies.

### G.1   Effect of Transformer Decoder

**Settings**   We conduct evaluations on a single Kmeans-constructed tree with 4 branches, and replace the Transformer Decoder with the RNN and CNN model. The hidden size and embedding size of RNN and CNN are set to be 96. Additionally, the kernel size of CNN is set to be 3.

**Results** The results are summarized in Table 9. We can see that the Transformer Decoder always performs best and the performance gaps between them are remarkable. This verifies the superiority of the Transformer Decoder for sequence generation tasks in our model compared to CNN and RNN.

Table 9: Effect of Transformer Decoder

| Decoder | NDCG@20 | | | |
|---|---|---|---|---|
| | Amazon | MIND | Gowalla | Yelp |
| Transformer | **0.1912** | **0.6935** | **0.2795** | **0.2269** |
| RNN | 0.1208 | 0.6912 | 0.2286 | 0.1984 |
| CNN | 0.1218 | 0.6933 | 0.1758 | 0.1691 |

## G.2 Effect of Joint Training

**Settings** To reveal the impact of joint representation learning, we include another experiment, where the joint representation learning is disabled. To be specific, we initialize the index parameter, i.e., the embedding of each branch, as the average of all the item embeddings within the corresponding branches. The parameters of the index are fixed during the training process, so that only the parameters of Encoder and Decoder are learnable. Such a variant is short for "Fixed Index" in our experiment.

**Results** The results are summarized in Table 10. The performance of fixed index is worse than the original one, which indicates the removal of joint representation learning indeed degrades the capability of our model. In other words, joint training is indeed important and necessary.

Table 10: Effect of Joint Training

| Index | NDCG@20 | | | |
|---|---|---|---|---|
| | Amazon | MIND | Gowalla | Yelp |
| Joint Training | **0.1912** | **0.6935** | **0.2795** | **0.2269** |
| Fixed Index | 0.1429 | 0.6891 | 0.1604 | 0.1588 |