# OpenReview forum: "Recommender Forest for Efficient Retrieval"
_NeurIPS.cc/2022/Conference — NeurIPS 2022 Accept_

### Official Review · Reviewer_ep4c · 2022-07-07

**Rating:** 6
**Confidence:** 4
**Soundness:** 3 good
**Presentation:** 3 good
**Contribution:** 3 good

**Summary:**

Approximate nearest neighbour search (ANNs) is used to retrieve top-N results for a user from a massive item set in personalized recommendation.
The paper creatively formulates the approximate nearest neighbour search (ANNs) problem into the sequence-to-sequence problem: the user profile and history is fed into the encoder of a Transformer while the top-N results are obtained by the beam-search of the decoder of that Transformer. To support this, the items are organized into a K-ary tree. To enhance the robustness, they construct several trees to form a forest.

**Questions:**

(1) What if they use other decoder instead of Transformer? like RNN or CNN-based decoder?

(2) Yes, RecForest significantly outperforms all efficient recommenders with indexes. But what contributes most to this improvement? the Transformer's power or the effectiveness of joint representation learning?

(3) Why index size of JTM and TDM is O(ND) since they are also tree-based methods as shown in Figure 1?

(4) The K-ary tree means the vocabulary of the decoder is K, right? In NLP, K is a large number like 30,000. But in your case, K should be a very small number like 2 if binary, right? Hence, I just wonder will such big difference post any training issues on Transformer?

(5) Based on (4), since K is a small number, is that necessary to share the "word" embeddings among the trees to save the memory?

(6) In real world recommendation, there are two stages: top-N retrieve and rerank. So this RecForest is for the top-N retrieve stage but not for the rerank stage, right? Since the performance is still worse than DIN in the experiments.

**Limitations:**

I don't see negative societal impact of their work,

**Strengths And Weaknesses:**

Strength:

(1) Hats off to the core idea "Given the hierarchical numbering of the items, the recommendation turns out to be a sequence-to-sequence problem: based on the encoded user information, paths to the most preferable items are progressively decoded via beam search, from which the top-n recommendation is made. "  Hence, the problem of ANN and item embedding learning are merged and solved in one system.

Is this the first paper that using such beam search way based on Transformer decoder to generate the Top-N results for recommendation?

(2) The experiments show RecForest significantly outperforms all efficient recommenders with indexes on all datasets w.r.t NDCG@20 and NDCG@40.

Weakness:
I don't see obvious weakness but hope the authors can carefully answer my questions presented in the following section because I feel some ablation study might be missing.

---

> ### Author Response · Authors · 2022-08-02
> **Responses to Reviewer ep4c**
>
> Many thanks for your positive comments and helpful questions! The detailed responses to your questions are listed below:
>
> **S1: Is this the first paper that using such beam search way based on Transformer decoder to generate the Top-N results for recommendation?**
>
> Yes. To the best of our knowledge, it is the first paper that transforms the recommendation task into a seq2seq task and uses the beam search to generate the sequence.
>
> **Q1: What if they use other decoders instead of Transformer? like RNN or CNN-based decoder?**
>
> Many thanks for your questions! We replace the Transformer Decoder with RNN and CNN model, the results are as follows. We can see that the model with Transformer Decoder always performs best.
>
>   |  (NDCG@20)  | Amazon |  MIND  | Gowalla |  Yelp  |
>   | :---------: | :----: | :----: | :-----: | :----: |
>   | Transformer | 0.1912 | 0.6935 | 0.2795  | 0.2269 |
>   |     RNN     | 0.1208 | 0.6912 | 0.2286  | 0.1984 |
>   |     CNN     | 0.1218 | 0.6933 | 0.1758  | 0.1691 |
>
> **Q2: What contributes most to the improvement? the Transformer's power or the effectiveness of joint representation learning?**
>
> According to the above experimental results, Transformer indeed contributes to the recommendation accuracy. To further analyze the impact of joint training, we introduce another experiment, where the joint training is disabled. To be specific, we initialized the index parameter, i.e., the embedding of each branch, as the average of all the item embeddings within the corresponding branch. The index parameters are fixed afterward, such that the training will only update the remaining parameters within the recommender. Such a variation is referred as "Fixed Index" in our experiment. According to the experiment results presented below, the fixed index is worse than our original performance, which means the removal of joint training is unfavorable. In other words, the effectiveness of joint training can be reflected.
>
>   |     (NDCG@20)     | Amazon |  MIND  | Gowalla |  Yelp  |
>   | :---------------: | :----: | :----: | :-----: | :----: |
>   |     Original      | 0.1912 | 0.6935 | 0.2795  | 0.2269 |
>   | Fixed Index | 0.1429 | 0.6891 | 0.1604  | 0.1588 |
>
>
> **Q3: Why index size of JTM and TDM is O(ND) since they are also tree-based methods as shown in Figure 1?**
>
> This is because we have to store the embedding of dimension D for each internal/leaf node in trees of JTM and TDM. Therefore, the index size of JTM and TDM is O(ND). However, RecForest only stores branch embedding and computes the node representation with the transformer decoder on the fly, so the index size of RecForest is much smaller than JTM and TDM.
>
> **Q4:Will a small K post any training issues on Transformer?**
>
> Thanks for this insightful question! And the answer to your question is No. We test different values of K (see Table 4) and they are really small compared to K in NLP, but there is no issue happening during the training process.
>
> **Q5: Since K is a small number, is that necessary to share the "word" embeddings among the trees to save the memory?**
>
> Since the total number of branches within the tree is at the same magnitude as the number of items, sharing the embeddings of the “K branches” among different levels of trees saves a lot of model parameters. However, the branch embeddings are not shared among different trees, as shown in *Index Size* of Table 1. Sharing branch embedding among different trees leads to the degradation of retrieval accuracy according to our empirical observations.
>
> **Q6: Is it for the top-N retrieve stage but not for the rerank stage? (worse than DIN)**
>
> Our Reforest mainly targets on the retrieval of the top-N items, thanks to its high running efficiency and low memory consumption. At the same time, it is also OK to apply it as an end-to-end recommender, considering that it achieves comparable performance to DIN in many cases (as Table 2).

---

> > ### Comment · Reviewer_ep4c · 2022-08-05
> > **feedback**
> >
> > Thanks for the reply,
> >
> > Basically I feel good about this paper and their responses.
> >
> > Just please add these ablation study to the paper and hence make it stronger.
> >
> > And I would respectfully and nicely suggest the authors do not over-state this method as an end-to-end recommender, since it is highly rare to use end-to-end rather than multi-stage in real industrial world.

---

> > > ### Author Response · Authors · 2022-08-09
> > > **Thanks for your feedback**
> > >
> > > Thanks for your approval of our work and insightful suggestions.
> > >
> > > Following your suggestions, we have included these ablation studies into the paper, which is now located in Sec. G of the appendices of the updated submission.

---

### Official Review · Reviewer_bwSz · 2022-07-12

**Rating:** 6
**Confidence:** 3
**Soundness:** 2 fair
**Presentation:** 3 good
**Contribution:** 1 poor

**Summary:**

The authors are interested in efficiently retrieving personalized items to users based on users' preferences. They focused on the efficacy aspect of the recommendation problem. The empirical results consist of several datasets with different ranges for # users, # items, and # interactions. They also included an analysis of performance on the quality of suggestions.


**Questions:**

The authors listed the following limitations of previous approaches to justify their framework:

- it is challenging to route to items located around the partition boundaries, given that the item set is hierarchically partitioned,
- the accuracy of the beam search can be limited by the routing decision, which is made without consideration of the previous trajectory,
- the tree-based index can be memory-consuming, given that the number of internal nodes is at the same magnitude as the leaf nodes,
- there is a high cost for the training stage of existing approaches when calling for the joint adaptation of the representation model and tree index, given that the tree structure needs to be repetitively updated.

It would be helpful to link the above statements/following features of RecForest and empirical results. For instance, a reference to which section contains the results that present the advantages to RecForest over baselines.

The authors correctly included the complexity analysis of essential times and index size. This helps understand the tradeoffs between previous literature and the proposed framework.


**Limitations:**

The authors need to highlight the limitations of their method, preferably contrasting it and previous approaches.


**Strengths And Weaknesses:**

The efficacy of recommender systems is often a neglected problem in tradition RecSys literature. Most of the works are interested in improve the quality of recommendations. However, this is a challenging and important problem, mainly, in practical settings.

It would be helpful to include statistical significance tests of the empirical results.

---

> ### Author Response · Authors · 2022-08-02
> **Responses to Reviewer bwSz**
>
> Many thanks for your helpful and detailed comments. The detailed responses to your questions are listed as follows:
>
> **W1:It would be helpful to include statistical significance tests of the empirical results.**
>
> Thank you for your suggestion! We conduct the significance tests between RecForest and the best index-based baseline (i.e. IPNSW in general) by repeating each method 5 times. We report the p-value of paired T-test as follows. In general, RecForest outperforms IPNSW in all cases and the superiority is significant in most cases. We need to point out that RecForest not just outperforms baselines in recommendation accuracy, but also beats the baselines in retrieval efficiency and memory cost.
>
> |               |  Movie  | Amazon  | Gowalla |  Tmall  |  MIND   |  Yelp   |
> | :-----------: | :-----: | :-----: | :-----: | :-----: | :-----: | :-----: |
> | p for NDCG@20 | 0.00001 | 0.00549 | 0.00003 | 0.00002 | 0.00306 | 0.00107 |
> | p for NDCG@40 |    0    | 0.10965 | 0.00003 | 0.00002 | 0.00018 | 0.00309 |
>
>
> **Q1: It would be helpful to link the above statements/following features of RecForest and empirical results.**
>
> Thanks for your question! We've added the linkage between our contributions to their experimental studies as follows. Besides, corresponding discussions in the Introduction (line 53-69) have also been revised accordingly.
>
> (1) Near-boundary item retrieval: Multiple trees are the key to solving this problem, so we conduct an ablation study on the Effect of Forest Construction (see section 3.6.1). The results in Figure 4 show that multiple trees achieve better results than a single tree, which confirms that a near-boundary item missed in one tree can be retrieved back on another tree.
>
> (2) Previous trajectory: The results in Table 3 show that our method performs better than previous methods  which merely take account of the current node (i.e., TDM and JTM).
>
> (3) Memory-consuming tree index: In Table 1, we analyze the space complexity of indexes. In Table 3, we list their real memory cost. Both indicate the superiority of RecForest in memory consumption to other indexes.
>
> (4) Joint training cost (update cost):  Due to the use of multiple trees, RecForest is much less sensitive to the partition of the item set. As a result, without any tree update, RecForest can perform remarkably better than TDM and JTM with repetitive tree updates, as shown in Table 3. Therefore, RecForest can avoid the repetitive adaption of the tree structure, saving a considerable portion of the training cost. In *Indexing Time* in Table1, we analyzed the indexing construction time for various indexing methods. The JTM costs the longest time, scaling linearly with the training size.
>
> **Limitation: The authors need to highlight the limitations of their method, preferably contrasting it and previous approaches.**
>
> Thanks for your insightful suggestions and we have presented them in the conclusion and future work. The higher efficient decoder method like non-autoregressive prediction can further improve the inference efficiency. In addition, we construct the tree based on Kmeans algorithm, but there may exist some better ways to capture the similarity between items so that a better tree index can be generated.

---

> > ### Comment · Reviewer_bwSz · 2022-08-09
> > **Thanks**
> >
> > The authors answered all my questions and addressed all my comments. Hence, I updated my overall rating.

---

### Official Review · Reviewer_L1cc · 2022-07-20

**Rating:** 8
**Confidence:** 4
**Soundness:** 3 good
**Presentation:** 3 good
**Contribution:** 3 good

**Summary:**

In this work, a tree-based recommendation method RecForest is proposed to improve the model efficiency of tree-based method. Also, the model training cost can be improved with the multiple K-ary trees. The retrieval of near-boundary items becomes more effective. Experiments are performed on several recommendation datasets to show the model effectiveness.

**Questions:**

Considering that the model evaluation tackles the sequential recommendation scenario, it would be better to further compare the new proposed RecForest approach with Transformer-based sequential recommender systems (e.g., Bert4Rec), or discuss the advantage of the new mode. The new method is also built over the Transformer architecture for sequence encoding.

What are the specific parameter settings of some compared baselines in the evaluation section?


**Limitations:**

Please find the limitations in the weakness part.

**Strengths And Weaknesses:**

Strengths:
1.The limitation of near-boundary item retrieval has been addressed in this work.
2.The ablation study is provided to investigate the effects of forest construction.
3.Several widely used experimented datasets are used for model evaluation.

Weaknesses:
1.In the evaluation section, it would be better to describe the detailed settings of some compared baselines, such as DIN and YoutubeDNN.
2. It would be better to discuss the potentials of the proposed method in tackling other types of recommendation scenarios, in addition to the sequential recommendation scenario.

---

> ### Author Response · Authors · 2022-08-02
> **Responses to Reviewer L1cc**
>
> Thank you for your insightful suggestions! We appreciate your encouragement and provide detailed responses to your comments below:
>
> **W1: The detailed settings of some compared baselines, such as DIN and YoutubeDNN.**
>
> Due to the space limitation, we briefly introduce the experimental settings in Section 3.3 of the main text. For more details, please refer to Appendix E.
>
> **W2: The potentials of the proposed method in tackling other types of recommendation scenarios.**
>
> Thanks for the suggestions! We have indeed conducted the same experiments in the non-sequential scenarios as in the sequential scenarios. And similar conclusions can be summarized from the experimental results obtained in non-sequential scenarios. All the results can be referred to in Appendix F.
>
> **Q1: Compare the new proposed RecForest approach with Transformer-based sequential recommender systems (e.g., Bert4Rec).**
>
> Thanks for your insightful questions! We additionally include two Transformer-based sequential models (e.g. SASRec and Bert4Rec), which are ranking models and can be utilized to recommend items by brute-force ranking all items. The results on all datasets are as follows.  RecForest can outperform both SASRec and Bert4Rec in most cases except on Gowalla. These results further validate the effectiveness of RecForest.
>
>
>
> |           | Movie       |         | Amazon    |         |
> | --------- | ----------- | ------- | --------- | ------- |
> |           | NDCG@20     | NDCG@40 | NDCG@20   | NDCG@40 |
> | SASRec    | 0.402       | 0.4394  | 0.1868    | 0.2104  |
> | BertRec   | 0.3841      | 0.4151  | 0.1241    | 0.1419  |
> | RecForest | 0.5580      | 0.5682  | 0.2339    | 0.2570  |
> |           | **Gowalla** |         | **Tmall** |         |
> |           | NDCG@20     | NDCG@40 | NDCG@20   | NDCG@40 |
> | SASRec    | 0.3955      | 0.403   | 0.0879    | 0.1026  |
> | BertRec   | 0.3626      | 0.3766  | 0.1057    | 0.1247  |
> | RecForest | 0.3783      | 0.3963  | 0.2059    | 0.2261  |
> |           | **MIND**    |         | **Yelp**  |         |
> |           | NDCG@20     | NDCG@40 | NDCG@20   | NDCG@40 |
> | SASRec    | 0.6652      | 0.6682  | 0.2309    | 0.2639  |
> | Bert4Rec  | 0.5024      | 0.5177  | 0.2478    | 0.2773  |
> | RecForest | 0.7583      | 0.7579  | 0.2766    | 0.3031  |
>
> **Q2: What are the specific parameter settings of some compared baselines in the evaluation section?**
>
> Please see the above response to W1.

---

> > ### Comment · Reviewer_L1cc · 2022-08-09
> > **Acknowledgement of author response**
> >
> > Thanks for clarifying the experimental settings. I would like to raise my evaluation score and vote for acceptance on this work.

---

### Meta-Review · Area_Chair_LtvT · 2022-08-28

**Recommendation:** Accept
**Confidence:** Certain

**Metareview:**

The paper introduces a method for top-n item recommendation based on approximate nearest neighbor search (ANN). The authors formulate ANN as a sequence to sequence problem, the input being the user profile and activity, and the output being the top-n recommendations. The focus of the paper is on the computational efficiency of the ANN process. The proposed method learns jointly a tree-based index for organizing the items and a transformer based decoder for the top-n recommendation. The index is composed of multiple trees. Experiments are performed on classical benchmarks.


The reviewers consider that this is an original contribution with a convincing experimental evaluation. The authors have added several complementary experiments, including additional baselines, during the rebuttal and answered satisfyingly to the reviewers’ comments and questions. All the reviewers recommend acceptance.


**Award:**

No

---

### Decision · Program_Chairs · 2022-09-14

Accept